# Posttraumatic Stress Symptoms and Attitudes toward the China Eastern Airlines Plane Crash in Transportation Students

**DOI:** 10.3390/ijerph191811400

**Published:** 2022-09-10

**Authors:** Lei Xia, Cheng Yang, Jiawei Wang, Lewei Liu, Yinghan Tian, Yi-lang Tang, Feng Jiang, Huanzhong Liu

**Affiliations:** 1Department of Psychiatry, Chaohu Hospital of Anhui Medical University, Hefei 238000, China; 2Department of Psychiatry, School of Mental Health and Psychological Sciences, Anhui Medical University, Hefei 230032, China; 3Department of Psychiatry, Anhui Psychiatric Center, Anhui Medical University, Hefei 230032, China; 4Department of Psychiatry, Bozhou People’s Hospital, Bozhou 236800, China; 5Department of Psychiatry and Behavioral Sciences, Emory University, Atlanta, GA 30322, USA; 6Atlanta Veterans Affairs Medical Center, Decatur, GA 30033, USA; 7School of International and Public Affairs, Shanghai Jiao Tong University, Shanghai 200030, China; 8Institute of Healthy Yangtze River Delta, Shanghai Jiao Tong University, Shanghai 200030, China

**Keywords:** plane crash, transportation students, posttraumatic stress symptoms, depression, indirect traumatization

## Abstract

On 21 March 2022, a China Eastern Airlines plane with 132 people on board crashed and all people are presumed dead. This study aimed to explore mental health symptoms and attitudes toward the plane crash among flight and train attendant students and the general public. A cross-sectional online survey was conducted two weeks after the plane crash. Mental health symptoms, including posttraumatic stress symptoms (PTSS), depressive, anxiety, and insomnia symptoms were assessed. A total of 494 participants were included, of which 183 were flight (*n* = 140) and train (*n* = 43) attendant students (aged 17.3 ± 1.7 years, 80.9% were female), and 311 were sampled from the general population (aged 26.7 ± 7.8 years, 62.1% were female). The prevalence of depressive, anxiety, and insomnia symptoms, and PTSS was 51.9%, 40.4%, 25.1%, and 12.6% in the transportation students, and 45.3%, 36.0%, 17.4%, and 4.2% in the general public sample, respectively. The students reported more frequent insomnia symptoms and PTSS than the general public sample. In the student group, compared with those without PTSS, those with PTSS reported significantly higher rates of depressive, anxiety, and insomnia symptoms. Two weeks after a plane crash, mental health symptoms are common in the general public and transportation students, with the latter being more likely to have PTSS symptoms. Our findings suggest the importance to identify risk groups when developing interventions after indirect exposure to traumatic events.

## 1. Introduction

### 1.1. The China Eastern Airlines Plane Crash

A China Eastern Airlines passenger plane with 132 people on board crashed on 21 March 2022, and all people are presumed dead [1]. As with many other disasters involving planes or trains, the constant news media and social media coverage of the disaster and the victims may have negative impacts on the mental health of those who are not directly involved in the disaster [2].

### 1.2. PTSD or PTSS Following Disasters

The impact of natural or human-made disasters on the mental health of victims and the general public has received considerable research attention. Post-traumatic stress disorder (PTSD) is one of the most frequently reported mental health consequences in the aftermath of disasters [3], which is characterized by a series of post-traumatic stress symptoms (PTSS), such as intrusion, active avoidance, negative cognitions, and emotions, and marked alterations in arousal and reactivity [4]. Many previous studies found a higher rate of PTSD or PTSS following natural disasters (e.g., tsunamis, floods, and earthquakes) [5,6,7] and human-made disasters (e.g., war and terrorist attacks) [3].

It is important to understand that the psychological responses depend greatly on the nature of traumatic events (type and intensity of exposure), sampling procedures and populations (victims, rescue workers, and the general population), and time-frames for follow-ups [3,8]. For example, a previous study compared the psychological impact of two technological disasters and found that people exposed to an aircraft crash experienced more symptoms of intrusion and avoidance than those exposed to a train collision [8]. Moreover, these symptoms were more frequent among people who had experienced high exposure to traumatic events than in the low/medium exposure group [8]. A systematic review of PTSD following disasters revealed that the prevalence of PTSD is higher among victims who were directly exposed to disasters, lower among rescue workers, and even lower in the general population [3]. Furthermore, a general decline could be found in the prevalence of PTSD over time after exposure to traumatic events [3].

### 1.3. Indirect Exposure to Traumatic Events and Impact of Media Coverage

The psychological effects of major trauma are not limited to those who have directly experienced them. A national study found that 17% of Americans outside of New York City reported PTSS two months after the September 11 attacks [9]. A recent meta-analysis found the pooled prevalence of PTSS following large-scale pandemics (e.g., sudden acute respiratory syndrome [SARS], H1N1, Ebola, and coronavirus disease 2019 [COVID-19]) was 19.3% among the general public, who were not infected or even not exposed to these diseases [10].

Similarly, major transportation (airplanes, trains, etc.) accidents affect not only direct victims, but also indirect victims, even the general public. For the direct victims of transportation accidents, a study showed that 6 months after the Bijlmermeer plane crash in the Netherlands in 1992, 26% of victims from the most severely damaged buildings were found to have a diagnosis of PTSD based on the 3rd edition of the Diagnostic and Statistical Manual of Mental Disorder (DSM-III) [10]. Studies found that almost all survivors of a plane crash reported PTSS, such as hypervigilance, startled reaction, and difficulty in sleeping [11], and those from train accidents were likely to suffer from depression, anxiety, irritability, insomnia, and nightmares [12,13]. For the indirect victims of transportation accidents, a previous study of community residents found that those who lived further away from the train crash site also reported considerable levels of intrusion or avoidance symptoms although they are not injured [8]. Another study found that after an airplane crash in Ukraine that killed 196 Dutch people, Dutch adults sampled from the general population reported acute somatic and affective symptoms in the days directly after the crash, although they lived 2600 km away from the crash site [2]. More media exposure was found to be associated with more negative affect [2]. The findings suggest that with vicarious or indirect exposure via radio, television, and social media, disasters may impact people geographically far away.

Sensational, graphic, or selective media coverage of catastrophic accidents can present an overly negative and biased reality, causing negative emotional reactions (such as fear) in readers or viewers [14]. On the other hand, humans are known to be irrational and emotionally affected regarding risk assessment, and the media coverage often distorts audiences’ risk perceptions of flying and may affect their choice of transportation means [14].

### 1.4. Purpose of This Study

Compared with the general public with similar (indirect) exposure, people who feel psychologically connected to the victims (either due to personal or professional background, or because the accident is on the same route as one’s traveling habits, etc.) may be more severely impacted. However, data are scarce on the psychological effects of plane crashes on the general public or people with a career future in transportation. This current study aimed to explore PTSS and other mental health symptoms, such as depressive, anxiety, insomnia, and fatigue symptoms among flight and train attendant students and the general public two weeks after the China Eastern Airlines plane crash, and attitudes toward the plane crash. We also examined the clinical factors associated with PTSS.

## 2. Materials and Methods

### 2.1. Study Design and Participants

An anonymous cross-sectional online survey was conducted between 4 April 2022, and 10 April 2022, two weeks after the March 21 plane crash occurred. All students from two programs of flight and train attendants at a comprehensive college in northern Anhui Province, China were invited to participate in the survey. During the same period, users of WeChat, a popular social network app in China, were also invited to complete the anonymous survey. This study was approved by the Ethics Committee of Chaohu Hospital, Anhui Medical University. Electronic informed consent was received before the respondents answered questions.

### 2.2. Measuring Instruments

#### 2.2.1. Demographic Characteristics

The questionnaire used in this study consisted of three parts, including socio-demographic information, assessments of mental health symptoms (see below), and experience of and attitudes toward the plane crash. Detailed instructions were provided before each section. Socio-demographic data including occupations, age (years), gender (male or female), and marital status (single, married, divorced, or widowed) were collected. The occupations of participants in the general public group were not limited and may include healthcare workers, enterprise or institution workers, teachers or students, or other occupations, such as merchants, freelancers, etc. [15].

#### 2.2.2. Mental Health Symptoms

##### Depressive symptoms

The Patient Health Questionnaire-9 (PHQ-9) was used to assess depressive symptoms in the past 2 weeks [16]. The Chinese version of PHQ-9 has been validated with good psychometric properties [17,18]. It consists of nine items with an individual score ranging from “0 = not at all” to “3 = nearly every day”. In this study, participants with a total PHQ-9 score of ≥5 were considered as “having depressive symptoms” [16].

##### Anxiety symptoms

The Chinese version of Generalized Anxiety Disorder-7 (GAD-7) was used to assess anxiety symptoms [19,20,21]. The seven items were also rated on a 4-point Likert scale ranging from “0” to “3”, with a higher total score indicating more severe anxiety symptoms. In this study, a cut-off value of ≥5 was used for identifying with or without anxiety symptoms [19].

##### Insomnia symptoms

The Chinese version of the Insomnia Severity Index (ISI) was used to assess insomnia symptoms [22,23,24]. It is a self-rated questionnaire, and it has 7 items which were scored on a 4-point scale from “0 = none” to “4 = very severe”. In this study, a cut-off value of ≥8 for the ISI total score was used for identifying insomnia symptoms [22].

##### Fatigue symptoms

A numeric rating scale (NRS) was used to measure fatigue symptoms [25,26]. It is a single item scoring from “0 = no fatigue” to “10 = unbearable fatigue”.

##### Posttraumatic stress symptoms

The 17-item PTSD Checklist-Civilian Version (PCL-C) [27] was used to assess the PTSS following the recent China Eastern Airlines plane crash. This scale has been previously validated in a Chinese sample [28]. Each item of the PCL-C was scored on a 5-point Likert scale from “1 = not at all” to “5 = extremely”. The PCL-C total score ranged from 17 to 85, with a score of ≥38 reflecting “having clinically PTSS” [29,30,31]. The time frame for the PTSS assessments was flexible to fit specific needs, and it was set as “the past two weeks” in this study [27].

In this study, the Cronbach’s alpha coefficient of PHQ-9, GAD-7, ISI, and PCL-C was 0.913, 0.934, 0.905, and 0.951, respectively.

#### 2.2.3. Experience and Attitudes

The experience (Q1–Q4) of, and attitudes (Q5–Q8) toward the recent China Eastern Airlines plane crash were evaluated using the following 8 items: Q1. “Do you think flying is a safe way of transportation?” (Answer: “Yes. It is a safe way”; or “No. It is an unsafe (dangerous) way.”). Q2. “Which means of transportation do you think are relatively safe that you will choose (Please select at least one option)”? (Answer: “Railway”; “Foot”; “Air”; “Car”; “Bicycle”; “Bus”; “Water”; “Van”; and “Motorcycle”). Q3. How often do you travel by airplane? (Answer: ”Never”; “A few times a year”; “A few times a month or more”). Q4. “Have you or any families/friends experienced any sudden or unexpected events when flying?” (Answer: “Yes” or “No”). Q5. “Does the China Eastern Airlines plane crash have any impact on your choice of air travel?” (Answer: “Yes. I will not choose to fly in the future”; “Yes. I will reduce the frequency of air travel”; “No. I will choose transportation as needed”; “No. I will give preference to air travel if the conditions permit”). Q6: “Times spent thinking about the plane crash (Including reading news on Wechat, watching TV, or listening to Radio, discussing the information about the plane crash with family and friends, etc.)” (Answer: “<1 h per day”; “1–2 h per day”; “≥3 h per day”) [15]; Q7. “Have you been physically and psychologically affected by the China Eastern Airlines plane crash?” (Answer: “No”, “Mild”, “Moderate”, or “Severe”). Q8. “Do you need professional psychological assistance?” (Answer: “Yes” or “No”).

### 2.3. Data Analysis

We conducted the data analyses using IBM SPSS (Version 23.0). Continuous and categorical variables were described as mean ± standard deviation (SD) and frequency (%), respectively. We compared the differences between groups using the chi-square test for categorical variables, *t*-test, and Mann-Whitney U-test for continuous variables as appropriate. When we compared the rates of depressive, anxiety, and insomnia symptoms, and PTSS between groups, logistic regression analyses were conducted to control for demographic factors, with group membership (attendant students vs. the general sample, or students with PTSS vs. without PTSS) as an independent variable using the “Enter” method. Similarly, when we compared the fatigue and PCL-C scores between groups, analysis of covariance (ANCOVA) was used to control for demographic factors with group membership as a fixed factor. Correlation coefficients between PCL-C score and PHQ-9, GAD-7, ISI, and fatigue scores were calculated for both the Pearson correlation and Spearman’s correlation. Since there were only 13 (4.2%) individuals with PTSS in the general sample, the comparison between participants with PTSS and those without PTSS was not performed. In this study, the sample size was calculated using PASS version 11.0. The PTSS rates in students and the general population were estimated to be 15% [31] and 5% [32], with a group difference of approximately 10%. Using the following relevant parameters of α = 0.05, 1−β = 0.90, and R = 2.0, the calculated sample size of the two groups was 135 and 270, respectively. The significance level of data analyses was set at *p* < 0.05.

## 3. Results

### 3.1. Participant Characteristics

A total of 223 transportation students were invited to participate in this survey, 183 (response rate = 82.1%) completed the assessments and were included in the final analysis. There were 140 flight and 43 train attendant students in the student group. The mean age of this group was 17.3 ± 1.7 years. Most of them were female (80.9%) and single (97.8%). The general public sample (*n* = 311) consisted of 147 (47.3%) healthcare workers, 81 (26.0%) teachers or students, 43 (13.8%) enterprise or institution workers, and 40 (12.9%) with miscellaneous occupations. The mean age was 26.7 ± 7.8 years. More than half (62.1%) were female and one-third (32.5%) were married (Table 1).

### 3.2. Mental Health Symptoms

The prevalence of depressive, anxiety, and insomnia symptoms, and PTSS in the attendant students was 51.9% (95% confidence interval [CI]: 44.6–59.2%), 40.4% (95% CI: 33.3–47.6%), 25.1% (95% CI: 18.8–31.5%), and 12.6% (95% CI: 7.7–17.4%), respectively (Table 2 and Appendix A). The prevalence rates in the general sample were 45.3% (95% CI: 39.8–50.9%), 36.0% (95% CI: 30.6–41.4%), 17.4% (95% CI: 13.1–21.6%), and 4.2% (95%CI: 1.9–6.4%). The attendant students reported more frequent insomnia symptoms (25.1% vs. 17.4%, *p* = 0.038) and PTSS (12.6% vs. 4.2%, *p* = 0.001), and had a higher PCL-C total score than the general population (26.3 vs. 22.6, *p* < 0.001). No significant difference was found in fatigue scores between the two groups (*p* = 0.105). The prevalence rates of clinical symptoms by group, gender, and severity were shown in Appendix A.

ANCOVA showed that the difference in the PCL-C total score (F(_1, 494)_ = 6.943, *p* = 0.009) between the two groups remained significant after controlling for age, gender, and marital status. However, after adjusting for demographic variables, the group differences in insomnia symptoms (odds ratio [OR] = 1.428, 95% CI: 0.808–2.526, *p* = 0.220) and PTSS (OR = 1.278, 95% CI: 0.522–3.128, *p* = 0.592) were not significant.

### 3.3. Experience and Attitudes

Overall, about 30% of respondents believe that flying is an unsafe or dangerous way of transportation (Table 3). The preferred means of transportation by frequency (%) reported by the whole sample were: railway (71.7), foot (55.1), air (36.6), car (28.9), bicycle (28.5), bus (25.7), water (8.1), van (3.8), and motorcycle (3.4). Sixteen (3.2%) respondents reported that they or their families/friends had experienced sudden/unexpected events during flying. Nearly half answered that they would not choose to fly (17.0%) or reduce the frequency of air travel (27.1%) in the future. More than two-thirds (69.8%) reported the perception of mild to severe physical and psychological effects of the plane crash, and 11.5% reported the perception of the need for psychological assistance. We also found that the attendant students spent more time ruminating about the plane crash (*p* < 0.001), and had a higher proportion reporting a need for psychological assistance (19.7%) than the general public (6.8%, *p* < 0.001) (Table 3).

### 3.4. Comparison between Attendant Students with and without PTSS

Table 4 shows the comparison between attendant students with (n = 23) and without PTSS (n = 160). Compared with the students without PTSS, those with PTSS reported higher rates of depressive (95.7% vs. 45.6%), anxiety (91.3% vs. 33.1%), and insomnia (73.9% vs. 18.1%) symptoms (all *p* < 0.001), and have a higher fatigue score (5.3 vs. 2.7, *p* < 0.001). After controlling for age, gender, marital status, and category of students (flight or train), the differences in the prevalence of depressive (OR = 28.632, 95%CI: 3.731–219.701, *p* < 0.001), anxiety (OR = 24.128, 95%CI: 5.279–110.269, *p* < 0.001), and insomnia symptoms (OR = 15.006, 95%CI: 5.163–43.607, *p* < 0.001), and fatigue score (F_(1, 183)_ = 30.201, *p* < 0.001) between the two groups remained significant.

In addition, the students with PTSS reported more frequent experience of sudden/unexpected events during flying (17.4% vs. 3.8%, *p* = 0.024), and had a higher proportion of the perception of moderate to severe physical and psychological effects by the plane crash (56.5% vs. 12.5%, *p* < 0.001).

### 3.5. Correlations among PHQ-9, GAD-7, ISI, Fatigue, and PCL-C Scores

Figure 1 shows a linear correlation between PCL-C score and PHQ-9, GAD-7, ISI, and fatigue scores, respectively, among flight and train attendant students. PCL-C score was positively correlated with PHQ-9 score (r = 0.668 for Pearson’s correlation; r = 0.555 for Spearman’s correlation, same hereafter), GAD-7 score (r = 0.698; r = 0.630), ISI score (r = 0.682; r = 0.571), and fatigue score (r = 0.463; r = 0.433) (all *p* < 0.001). The correlations between PHQ-9, GAD-7, ISI, fatigue, and PCL-C scores were similar in the general sample (Appendix A).

## 4. Discussion

### 4.1. Mental Health Symptoms and PTSS

As this was probably the first study to examine PTSS and other mental health symptoms two weeks following the China Eastern Airlines plane crash, our study provides a unique perspective on how a recent aircrash may affect transportation students and the general public. In this study, we found high rates of depressive, anxiety, and insomnia symptoms in both attendant students (51.9%, 40.4%, and 25.1%) and the general sample (45.3%, 36.0%, and 17.4%). Studies suggested that COVID-19 can be interpreted as a collective traumatic event that may lead to mental health symptoms and PTSS [33]. As a comparison, a recent large-sample survey using the same assessment tools for these symptoms reported relatively lower rates of depressive and anxiety symptoms, and a slightly higher rate of insomnia symptoms (27.9%, 31.6%, and 29.2%) in the general population during the COVID-19 pandemic [34]. These findings indicate that people are experiencing significant psychological distress, which may be a complex and comprehensive response to both the COVID-19 pandemic and the fresh trauma event of the plane crash instead of either one.

In this study, 12.6% of attendant students and 4.2% of the general public sample reported PTSS after the plane crash. The PTSS prevalence rate in the students surveyed is about five times that in a sample of home-quarantined university students (2.7%) [29] and is close to that among youth (14–35 years old) (14.4%) [31] in two recent studies using the same criterion for identifying PTSS, while the rate in the general sample is close to that in the Chinese public (4.6%) after the COVID-19 outbreak [32]. The results suggest that flight and train attendant students may be more vulnerable to PTSS after aircraft accidents. Furthermore, a significantly higher proportion of transportation students reported the need for psychological assistance than the general public (19.7% vs. 6.8%). These findings are likely due to the “psychological connections” between the transportation students with the crew members in the aircrash. These students are likely to identify themselves with the air crew and feel personally connected. These mental health symptoms, plus the perceived need for psychological assistance, suggest the need for early intervention, as some of the symptoms may persist and develop into clinical PTSD.

### 4.2. Experience and Attitudes

Concerning experience of and attitudes toward the plane crash, we found that a significant proportion (30%) of respondents held a distrustful view of flying, and nearly half responded that they would not choose to fly or reduce the frequency of air travel in the future. Plane crashes are very rare. According to statistics, there were 0.6 fatal accidents per 1 million flights worldwide and 0.4 fatal accidents per million flight hours during the decade from 2002 to 2011 [35]. In addition, we found that 37.2% of attendant students spent at least 1 h per day ruminating about the plane crash, which is much higher than the percentage of the general sample (16.1%). This may be one reason for the higher prevalence of PTSS among the students. The media bring people convenient information, but they often prefer to provide negative news, which is often more eye-catching. Therefore, there is a need to enhance the proper understanding of aircraft accidents and avoid over-focusing on negative reports, thereby reducing the appearance of fear and negative emotions.

### 4.3. Factors Associated with PTSS

In addition, we found that the students with PTSS reported more depressive, anxiety, insomnia, and fatigue symptoms than those without PTSS. Correlation analyses suggested the severity of PTSS was positively correlated with the severity of depressive, anxiety, insomnia, and fatigue symptoms in both samples. These validate the previous finding that emotional regulation is impaired in patients with PTSD [36,37]. For example, an imaging study found that patients with PTSD had increased activity in emotion-related brain regions than controls when they were exposed to negative distractors [36]. Another longitudinal study found that youths with post-earthquake PTSD had deficits in the emotional control domain of executive functioning compared to those without PTSD who were exposed to the same trauma event [36]. In clinical practice, it is possible that the relationship between PTSS and other mental health symptoms (such as depression, anxiety, and insomnia) is bi-directional and clinicians need to be aware of this.

### 4.4. Vicarious Traumatization Related to the Plane Crash?

As the public actively or passively follows the progress of a plane crash through traditional media and social media, people often have a strong sense of vicariousness accompanied by intense emotional experiences. Are these psychological effects vicarious traumatization caused by the plane crash? The initial definition of vicarious traumatization referred to the psychological consequences for therapists of exposure to clients’ traumatic experiences [38,39]. Recently, its application has been extended to several major disasters after which rescuers, health care workers, and the general public experience mental health symptoms similar to PTSS [40,41]. Interestingly, a recent study demonstrated that, as compared with frontline nurses, the general public and non-frontline nurses were experiencing a higher level of vicarious traumatization caused by the COVID-19 pandemic [41]. In any case, there is a need to pay more attention to and address the psychological impact of people indirectly exposed to such major trauma.

Learning to cope with disasters in the future is a top priority for public mental health. Under the unprecedented situation of the COVID-19 pandemic, the demand for mental health services is increasing, but the availability is greatly reduced in many countries, as well as in China [42,43]. Improving the ability to respond to disasters requires policymakers to be able to plan and allocate resources wisely, promote mental health, and enhance resilience to mitigate the negative effects of disasters. Specifically, for the vulnerable groups, such as students, mental health training programs in schools and crisis intervention centers may provide non-professional and professional psychological assistance.

### 4.5. Limitations

Several limitations in this study should be noted. First, we only surveyed flight and train attendant students from one college and the findings may not be generalizable to all transportation students in China. Second, we did not collect data before the accident. Without baseline data, we were also unable to compare these symptoms before and after the plane crash. Third, some relevant factors, such as prior exposure to trauma (such as accidents), the actual extent of media exposure related to the crash, and exposure to COVID-19 and its restrictive measures were not assessed. Fourth, to improve the timeliness of this survey and to avoid any risk of transmission of COVID-19, we used a Web-based data collection method and several self-reported scales (PCL, PHQ-9, GAD-7, and ISI) instead of clinical interview diagnoses for PTSD, depression, anxiety, and insomnia, although these scales have previously been validated in the Chinese sample. Finally, the timing of the survey is important, which was merely two weeks after the accident. Although we found high rates of mental health symptoms, and negative attitudes toward flying in the student sample and the general public sample, these symptoms and attitudes are likely to improve or change quickly over time [2].

## 5. Conclusions

In conclusion, a considerable number of attendant students and the general public experienced depressive, anxiety, and insomnia symptoms two weeks after the China Eastern Airlines plane crash. Compared with the general public sample, the students reported more PTSS and the need for psychological assistance, supporting the role of psychological connection in possible vicarious traumatization. Targeted screening and interventions need to be available for these students, and other people who share similar personal or professional backgrounds with the victims. For example, mental health training programs in schools and crisis intervention centers established in psychiatric settings may provide psychological assistance and support. Meanwhile, improving the ability to respond to such disasters requires policymakers to be able to better plan and allocate resources, promote mental health, and enhance resilience to mitigate the negative effects of disasters.

## Figures and Tables

**Figure 1 ijerph-19-11400-f001:**
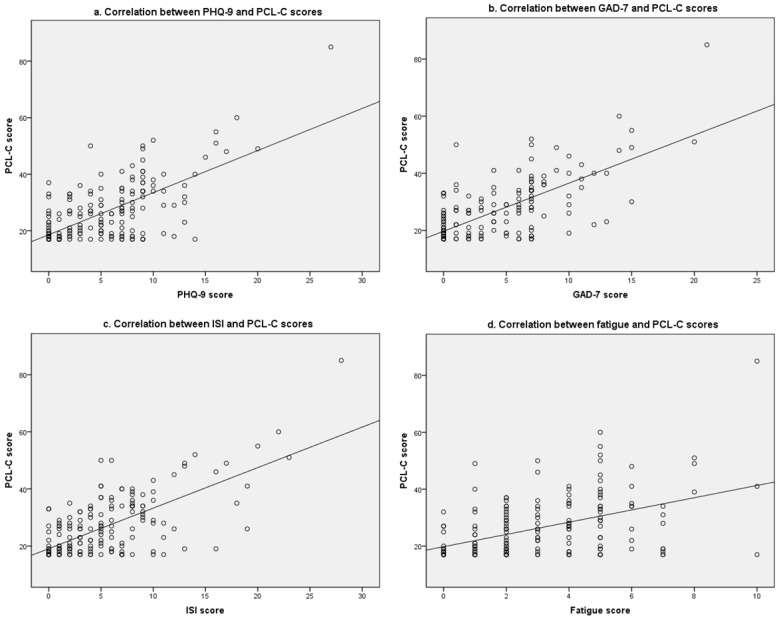
Scatter plots of the linear correlations between PHQ-9, GAD-7, ISI, fatigue, and PCL-C scores among flight and train attendant students.

**Table 1 ijerph-19-11400-t001:** Demographic characteristics of participants in this study.

Characteristics	Total (*n* = 494)	Attendant Students (*n* = 183)	General Public (*n* = 311)	t/χ^2^	*p*
Age (years, Mean ± SD)	23.2 ± 7.7	17.3 ± 1.7	26.7 ± 7.8	−20.432	<0.001
Gender (%)				19.080	<0.001
Male	153 (31.0)	35 (19.1)	118 (37.9)		
Female	341 (69.0)	148 (80.9)	193 (62.1)		
Marital status (%)					
Single	387 (78.3)	179 (97.8)	208 (66.9)	64.979	<0.001
Married	105 (21.3)	4 (2.2)	101 (32.5)		
Divorced or widowed	2 (0.4)	0	2 (0.6)		

**Table 2 ijerph-19-11400-t002:** Mental health symptoms of participants.

Variables	Total (*n* = 494)	Attendant Students (*n* = 183)	General Public (*n* = 311)	χ^2^/Z	*p*
Depressive symptoms (%)				1.996	0.158
Yes	236 (47.8)	95 (51.9)	141 (45.3)		
No	258 (52.2)	88 (48.1)	170 (54.7)		
Anxiety symptoms (%)					
Yes	186 (37.7)	74 (40.4)	112 (36.0)	0.961	0.327
No	308 (62.3)	109 (59.6)	199 (64.0)		
Insomnia symptoms (%)				4.312	**0.038**
Yes	100 (20.2)	46 (25.1)	54 (17.4)		
No	394 (79.8)	137 (74.9)	257 (82.6)		
Fatigue score (Mean ± SD)	3.2 ± 2.2	3.0 ± 2.3	3.3 ± 2.2	−1.622 ^1^	0.105
PCL-C score (Mean ± SD)	24.0 ± 9.0	26.3 ± 10.5	22.6 ± 7.7	−3.880 ^1^	**<0.001**
PTSS ^2^ (%)				11.998	**0.001**
Yes	36 (7.3)	23 (12.6)	13 (4.2)		
No	458 (92.7)	160 (87.4)	298 (95.8)		

^1^ Mann-Whitney U tests; Bolded value: <0.05; ^2^ PTSS, post-traumatic stress symptoms.

**Table 3 ijerph-19-11400-t003:** Experiences and attitudes toward the plane crash among participants.

Questions	Total (*n* = 494)	Attendant Students (*n* = 183)	General Public (*n* = 311)	χ^2^	*p*
1. Do you think flying is a safe way of transportation?				1.779	0.182
Yes. It is a safe way.	347 (70.2)	122 (66.7)	225 (72.3)		
No. It is an unsafe (dangerous) way.	147 (29.8)	61 (33.3)	86 (27.7)		
2. Which means of transportation do you think are relatively safe that you will choose (multiple choices)?					
Railway	354 (71.7)	99 (54.1)	255 (82.0)	44.143	**<0.001**
On foot	272 (55.1)	112 (61.2)	160 (51.4)	4.431	**0.035**
Air	181 (36.6)	56 (30.6)	125 (40.2)	4.566	**0.033**
Car	143 (28.9)	56 (30.6)	87 (28.0)	0.387	0.534
Bicycle	141 (28.5)	59 (32.2)	82 (26.4)	1.949	0.163
Bus	127 (25.7)	57 (31.1)	70 (22.5)	4.502	**0.034**
Water	40 (8.1)	17 (9.3)	23 (7.4)	0.555	0.456
Van	19 (3.8)	10 (5.5)	9 (2.9)	2.059	0.151
Motorcycle	17 (3.4)	9 (4.9)	8 (2.6)	1.908	0.167
3. How often do you travel by airplane?				29.406	**<0.001**
Never	341 (69.0)	153 (83.6)	188 (60.5)		
A few times a year	140 (28.3)	26 (14.2)	114 (36.7)		
A few times a month or more	13 (2.6)	4 (2.2)	9 (2.9)		
4. Have you or any families/friends experienced any sudden or unexpected events when flying?				4.594	**0.032**
Yes	16 (3.2)	10 (5.5)	6 (1.9)		
No	478 (96.8)	173 (94.5)	305 (98.1)		
5. Does the China Eastern Airlines plane crash have any impact on your choice of air travel?				6.569	0.087
Yes. I will not choose to fly in the future.	84 (17.0)	34 (18.6)	50 (16.1)		
Yes. I will reduce the frequency of air travel.	134 (27.1)	48 (26.2)	86 (27.7)		
No. I will choose transportation as needed.	221 (44.7)	73 (39.9)	148 (47.6)		
No. I will give preference to air travel if the conditions permit.	55 (11.1)	28 (15.3)	27 (8.7)		
6. Times spent thinking about the plane crash					
<1 h per day	376 (76.1)	115 (62.8)	261 (83.9)	29.293	**<0.001**
1–2 h per day	99 (20.0)	55 (30.1)	44 (14.1)		
≥3 h per day	19 (3.8)	13 (7.1)	6 (1.9)		
7. Have you been physically and psychologically affected by the China Eastern Airlines plane crash?				6.745	0.080
No	149 (30.2)	45 (24.6)	104 (33.4)		
Mild	274 (55.5)	105 (57.4)	169 (54.3)		
Moderate	53 (10.7)	23 (12.6)	30 (9.6)		
Severe	18 (3.6)	10 (5.5)	8 (2.6)		
8. Do you need professional psychological assistance?				18.840	**<0.001**
Yes	57 (11.5)	36 (19.7)	21 (6.8)		
No	437 (88.5)	147 (80.3)	290 (93.2)		

Bolded value: <0.05.

**Table 4 ijerph-19-11400-t004:** Comparison between flight and train attendant students with PTSS and those without.

Variables	With PTSS (*n* = 23)	Without PTSS (*n* = 160)	t/Z/χ^2^	*p*
** *Characteristics* **				
Age (years, Mean ± SD)	16.8 ± 1.2	17.4 ± 1.8	0.495	0.173
Gender (%)			- ^2^	1.000
Male	4 (17.4)	31 (19.4)		
Female	19 (82.6)	129 (80.6)		
Marital status (%)			- ^2^	0.418
Single	22 (95.7)	157 (98.1)		
Married	1 (4.3)	3 (1.9)		
Category of students (%)			0.045	0.832
Flight attendant students	18 (78.3)	122 (76.2)		
Train attendant students	5 (21.7)	38 (23.8)		
** *Mental health symptoms* **				
Depressive symptoms (%)	22 (95.7)	73 (45.6)	20.161	**<0.001**
Anxiety symptoms (%)	21 (91.3)	53 (33.1)	28.260	**<0.001**
Insomnia symptoms (%)	17 (73.9)	29 (18.1)	33.259	**<0.001**
Fatigue score (Mean ± SD)	5.3 ± 2.3	2.7 ± 2.1	−4.651 ^1^	**<0.001**
***Experiences and attitudes*** (Number of participants with a response of “yes”, %)				
Q1. Thought that airplane is unsafe (dangerous).	6 (26.1)	55 (34.4)	0.622	0.430
Q2. Tendency to choose air travel.	7 (30.4)	49 (30.6)	0.0003	0.985
Q3. Flying a few times a year or more.	9 (39.1)	21 (13.1)	- ^2^	**0.004**
Q4. Experience of sudden or unexpected events when flying.	4 (17.4)	6 (3.8)	- ^2^	**0.024**
Q5. Negative choice of air travel after the China Eastern Airlines plane crash.	13 (56.5)	69 (43.1)	1.459	0.227
Q6. Times spent thinking about the plane crash per day ≥1 h.	10 (43.5)	58 (36.2)	0.450	0.502
Q7. Perception of *moderate to severe* physical and psychological effects by the plane crash.	13 (56.5)	20 (12.5)	- ^2^	**<0.001**
Q8. Perception of the need for psychological assistance.	8 (34.8)	28 (17.5)	- ^2^	0.087

^1^ Mann-Whitney U tests; ^2^ Fisher’s Exact Test; Bolded value: <0.05.

## Data Availability

The data used for this study are available from the corresponding authors upon reasonable request.

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
