# Peer review of "Posttraumatic Stress Symptoms and Attitudes toward the China Eastern Airlines Plane Crash in Transportation Students"

_ijerph, 2022, doi:10.3390/ijerph191811400_

Round 1
Reviewer 1 Report
Thank you for the opportunity to review this study entitled “Posttraumatic stress symptoms and attitudes toward the China Eastern Airlines plane crash in transportation students.” (ijerph-1898369).
The research aimed at exploring mental health symptoms and attitudes toward a plane crash (On March 21, 2022) among flight and train attendant students and the general 22 public. The study focused on posttraumatic stress symptoms (PTSS), depression, anxiety, and insomnia symptoms and involved a sample of 494 participants (183 were flight and train attendant students; 311 were from the general population).
In my opinion, the research topic is relevant, and the study is interesting. Parallelly, some issues need to be addressed before the paper will be suitable for publication.
1. In the abstract, the information about the sample should be deepened (N? Mean age and SD? Percentage of men and women?) for both the two groups (flight and train attendant students; general population) to provide a clear picture of what will be presented in the paper.
2. Please remove headings in the abstract, according to the IJERPH guidelines.
3. Please provide information on internal consistency in this sample for the used scales.
4. The practical implications of these findings should be described in more depth in the “Conclusion” section.
Best wishes
Author Response
Responses to Reviewer 1
Reviewer #1:
Summary: The research aimed at exploring mental health symptoms and attitudes toward a plane crash (On March 21, 2022) among flight and train attendant students and the general 22 public. The study focused on posttraumatic stress symptoms (PTSS), depression, anxiety, and insomnia symptoms and involved a sample of 494 participants (183 were flight and train attendant students; 311 were from the general population).
In my opinion, the research topic is relevant, and the study is interesting. Parallelly, some issues need to be addressed before the paper will be suitable for publication.I consider that the manuscript offers information that is useful and provides important insight into the context of mental health. The study offers valuable information that can be useful in our current reality due to the workload and pressure that the pandemic has exerted within hospitals and clinics. I consider the study to be publishable.
Question 1: In the abstract, the information about the sample should be deepened (N? Mean age and SD? Percentage of men and women?) for both the two groups (flight and train attendant students; general population) to provide a clear picture of what will be presented in the paper.
Response: Thank you for the comments. Based on these comments, we have added the details (the number of flight and train attendant students, respectively; mean age and SD; and percentage of females for the two groups) to the Abstract section.
Question 2: Please remove headings in the abstract, according to the IJERPH guidelines.
Response: Thank you. We agree with the comment. We have now deleted the headings in the Abstract section.
Question 3: Please provide information on internal consistency in this sample for the used scales.
Response: Excellent point. We have now added the data in the Methods section under the subtitle- 2.2. Measuring instruments, showing as “In this study, the Cronbach’s alpha coefficient of PHQ-9, GAD-7, ISI, and PCL-C was 0.913, 0.934, 0.905, and 0.951 respectively.”
Question 4: The practical implications of these findings should be described in more depth in the “Conclusion” section.
Response: Thank you. Based on this comment, we have added more implications of these findings to the Conclusion section, showing as “For example, mental health training programs in schools and crisis intervention centers established in psychiatric settings may provide psychological assistance and support. Meanwhile, improving the ability to respond to such disasters requires policymakers to be able to better plan and allocate resources, promote mental health, and enhance resilience to mitigate the negative effects of disasters.”

Reviewer 2 Report
Dear colleagues, I hope this message find you well.
Thank you for giving me the opportunity of reading the work “Posttraumatic stress symptoms and attitudes toward the China Eastern Airlines plane crash in transportation students”, it has been a very big pleasure to collaborate reviewing this manuscript. The topic of this paper is very interesting and it seems necessary to delve it. However, there are several questions to improve before to publish it. I would suggest some changes:
Title and abstract
· Abstract: I recommend to eliminate all the numbers included (except sample information). This information could be found at the main document.
Introductio
Dear colleagues, the structure of the introduction is not clear. I recommended to divide it into several subsections. For example, creating a specific subsection where describe each group of variables/factors involved. Moreover, aims should be described in a clearly and specific way.
Method
The information regarding this section was described correctly.
Results
· Have you assessed/controlled another variable in order to avoid interferences?
Discussion
· When you explain the signs of psychological distress and mental health as a result of COVID-19, it is necessary to add more data and references. I recommend you to add this paper recently published (https://doi.org/10.3390/ijerph18147422), which proposes COVID-19 pandemic as a PTSD.
· In my humble opinion, it could be useful to describe in more detail the practical and theoretical implications of this research. It would be useful they contextualize better the contribution within the framework of the issue explaining why the contribution is useful and enrich the impact.
Conclusions
· Nothing to add. Good job.
Author Response
Responses to Reviewer 2
Reviewer #2:
Summary: Thank you for giving me the opportunity of reading the work “Posttraumatic stress symptoms and attitudes toward the China Eastern Airlines plane crash in transportation students”, it has been a very big pleasure to collaborate reviewing this manuscript. The topic of this paper is very interesting and it seems necessary to delve it. However, there are several questions to improve before to publish it. I would suggest some changes:
Title and abstract
Abstract: I recommend to eliminate all the numbers included (except sample information). This information could be found at the main document.
Response: Thank you. Combined with the question of Reviewer 1, we have added the details about the samples, and deleted the numbers included in the Abstract.
Introduction
Dear colleagues, the structure of the introduction is not clear. I recommended to divide it into several subsections. For example, creating a specific subsection where describe each group of variables/factors involved. Moreover, aims should be described in a clearly and specific way.
Response: Thank you very much. We have now divided the Introduction into several subsections: 1.1. The China Eastern Airlines plane crash; 1.2. PTSD or PTSS following disasters and impact of media coverage; 1.3. ……
We also have revised the aims of this study, showing as “This current study aimed to explore PTSS and other mental health symptoms, such as depressive, anxiety, insomnia, and fatigue symptoms among flight and train attendant students and the general public two weeks after the China Eastern Airlines plane crash, and attitudes toward the plane crash. We also examined the clinical factors associated with PTSS.”
Methods
The information regarding this section was described correctly.
Response: Thank you.
Results
Have you assessed/controlled another variable in order to avoid interferences?
Response: Yes. We have included this point in the Methods section under the subtitle- 2.3. Data analysis, showing as “When we compared the rates of depressive, anxiety, and insomnia symptoms, and PTSS between groups, logistic regression analyses were conducted to control for demographic factors, with group membership (attendant students vs the general sample, or students with PTSS vs without PTSS) as an independent variable using the “Enter” method. Similarly, when we compared the fatigue and PCL-C scores between groups, analysis of covariance (ANCOVA) was used to control for demographic factors with group membership as a fixed factor.”
Discussion
When you explain the signs of psychological distress and mental health as a result of COVID-19, it is necessary to add more data and references. I recommend you to add this paper recently published (https://doi.org/10.3390/ijerph18147422), which proposes COVID-19 pandemic as a PTSD.
Response: Excellent point. We have included this important point to the Discussion section under the subtitle- 4.1. Mental health symptoms and PTSS and cited this reference, showing as “Studies suggested that COVID-19 can be interpreted as a collective traumatic event that may lead to mental health symptoms and PTSS [33].”
In my humble opinion, it could be useful to describe in more detail the practical and theoretical implications of this research. It would be useful they contextualize better the contribution within the framework of the issue explaining why the contribution is useful and enrich the impact.
Response: Thank you. Combined with the question of Reviewer 1, we have added more implications of the findings to the Discussion and Conclusion sections, showing as “Specifically for the vulnerable groups, like students, mental health training programs in schools and crisis intervention centers may provide non-professional and professional psychological assistance” and “For example, mental health training programs in schools and crisis intervention centers established in psychiatric settings may provide psychological assistance and support. Meanwhile, improving the ability to respond to such disasters requires policymakers to be able to better plan and allocate resources, promote mental health, and enhance resilience to mitigate the negative effects of disasters.”
Conclusions
Nothing to add. Good job.
Response: Thank you.

Reviewer 3 Report
First of all, thank you for allowing me to review the manuscript.
Mental Health is a MASSIVE field, and, we as a scientific community should work on improving people's access to healthy minds. Although I think it is a very well-developed manuscript and the research is well-approached, there are several aspects that I would like to comment on:
-Line 53-56 and human-made disasters (e.g. war and terrorist attacks) [3]. It is 53 important to understand that the psychological responses depend greatly on the nature 54 of trauma events (type and intensity of exposure), sampling procedures, and populations: This paragraph approaches many variables, even for an introduction. I advise the authors to give it more content and at least make it more comprehensible. For a regular reader may be complicated to understand why all of them are important to be mentioned in the introduction.
-Line 69 PTSD is based on the 3rd edition of the Diagnostic 69 and Statistical Manual of Mental Disorder (DSM-III): There is a DSM-V, please cite the symptoms based on the current one: https://dsm.psychiatryonline.org/doi/book/10.1176/appi.books.9780890425787
-Line 89 Compared with the general public with similar (indirect) exposure, people who feel: How do they expose themselves to the accident? Youtube, Twitter, Twitch, Other mass media, TV, and Radio….I know there are some limitations in China concerning those platforms or social media, or about using those platforms, but please specify it, we are pointing to an international audience and readers, they should know. If this is not possible to be disclosed at this point, should be included as a limitation.
-All the scales used have previously been validated in the Chinese sample, thus, providing high fidelity data with valuable statistic power. I honestly think this should be mentioned in the strength/limitations section, this is what gives power to the manuscript and its findings.
-Line 310: Please use moderate language and be cautious when making associations, especially being the first investigation on this topic as the authors recognized.
-Line 333: Where are the applications to practice? It´s all about policymakers? Do we, healthcare professionals have something to learn? To apply from this? What about lecturers? Are we going to improve, change, or modulate our teaching style, and skills to help or prevent this? Please, we investigate to improve people's life, address that. Should be at the end of the discussion.
-Line 345: I completely agree with this, the time of the survey is crucial, is the accident still have media coverage? Are still vivid images and remembrances of what happened? Memorials? Are families affected by interviews being shown on media? Should be explained in depth to understand and interpret the findings.
It may seem like many details to approach, but honestly, I think that if authors address them, the manuscript match journal criteria and provide interesting and applicable findings.
Author Response
Responses to Reviewer 3
Reviewer #3:
Summary: Mental Health is a MASSIVE field, and, we as a scientific community should work on improving people's access to healthy minds. Although I think it is a very well-developed manuscript and the research is well-approached, there are several aspects that I would like to comment on:
Question 1: -Line 53-56 and human-made disasters (e.g. war and terrorist attacks) [3]. It is important to understand that the psychological responses depend greatly on the nature of trauma events (type and intensity of exposure), sampling procedures, and populations: This paragraph approaches many variables, even for an introduction. I advise the authors to give it more content and at least make it more comprehensible. For a regular reader may be complicated to understand why all of them are important to be mentioned in the introduction.
Response: Thank you for the comments. Based on these comments, we have added more content to this paragraph, showing as “It is important to understand that the psychological responses depend greatly on the nature of traumatic events (type and intensity of exposure), sampling procedures and populations (victims, rescue workers, and the general population), and time-frames for follow-ups [3,8]. For example, a previous study compared the psychological impact of two technological disasters and found that people exposed to aircraft crash experienced more symptoms of intrusion and avoidance than those exposed to train collision [8]. Moreover, these symptoms were more frequent among people who had experienced high exposure to traumatic events than in the low/medium exposure group [8]. A systematic review of PTSD following disasters revealed that the prevalence of PTSD is higher among victims who were directly exposed to disasters, lower among rescue workers, and even lower in the general population [3]. Furthermore, a general decline could be found in the prevalence of PTSD over time after exposure to traumatic events [3].”
Question 2: -Line 69 PTSD is based on the 3rd edition of the Diagnostic 69 and Statistical Manual of Mental Disorder (DSM-III): There is a DSM-V, please cite the symptoms based on the current one: https://dsm.psychiatryonline.org/doi/book/10.1176/appi.books.9780890425787
Response: Thank you for your carefulness. The study was conducted after the Bijlmermeer plane crash in the Netherlands in 1992, and the diagnosis of PTSD was based on the DSM-III. So we could not cite the DSM-V in this part. However, we have already cited the symptoms of PTSD diagnosis based on DSM-V in the second paragraph of the Introduction, showing as “…which is characterized by a series of post-traumatic stress symptoms (PTSS), such as intrusion, active avoidance, negative cognitions, and emotions, and marked alterations in arousal and reactivity [4]”.
Question 3: -Line 89 Compared with the general public with similar (indirect) exposure, people who feel: How do they expose themselves to the accident? Youtube, Twitter, Twitch, Other mass media, TV, and Radio….I know there are some limitations in China concerning those platforms or social media, or about using those platforms, but please specify it, we are pointing to an international audience and readers, they should know. If this is not possible to be disclosed at this point, should be included as a limitation.
Response: Thank you. Participants in this study were indirectly exposed to the plane crash via reading news on Wechat, TV, or Radio, discussing the information with family and friends, etc. Therefore, we have included this important point to the Methods section under the subtitle- 2.2.3. Experience and attitudes, showing as “Times spent thinking about the plane crash (Including reading news on Wechat, watching TV, or listening to Radio, discussing the information about the plane crash with family and friends, etc.)”
Question 4: -All the scales used have previously been validated in the Chinese sample, thus, providing high fidelity data with valuable statistic power. I honestly think this should be mentioned in the strength/limitations section, this is what gives power to the manuscript and its findings.
Response: Thank you. Based on the comment, we have now added this point in the Limitations section, showing as “Fourth, to improve the timeliness of this survey and to avoid any risk of transmission of COVID-19, we used a Web-based data collection method and several self-reported scales (PCL, PHQ-9, GAD-7, and ISI) instead of clinical interview diagnoses for PTSD, depression, anxiety, and insomnia, although these scales have previously been validated in the Chinese sample.”
Question 5: -Line 310: Please use moderate language and be cautious when making associations, especially being the first investigation on this topic as the authors recognized.
Response: Thank you. We have reworded the sentence to weaken the tone: “In clinical practice, it is possible that the relationship between PTSS and other mental health symptoms (such as depression, anxiety, and insomnia) is bi-directional and clinicians need to be aware of this.”
Question 6: -Line 333: Where are the applications to practice? It´s all about policymakers? Do we, healthcare professionals have something to learn? To apply from this? What about lecturers? Are we going to improve, change, or modulate our teaching style, and skills to help or prevent this? Please, we investigate to improve people's life, address that. Should be at the end of the discussion.
Response: Thank you. We have now added more implications of the findings to the end of the Discussion sections, showing as “Specifically for the vulnerable groups, like students, mental health training programs in schools and crisis intervention centers may provide non-professional and professional psychological assistance.”
Question 7: -Line 345: I completely agree with this, the time of the survey is crucial, is the accident still have media coverage? Are still vivid images and remembrances of what happened? Memorials? Are families affected by interviews being shown on media? Should be explained in depth to understand and interpret the findings.
Response: Yes. The accident still had a lot of media coverage when this study was conducted. Question 6 could reflect, to some extent, the intensity of exposure to the plane crash through media coverage at the time of the investigation. Q6: “Times spent thinking about the plane crash (Including reading news on Wechat, watching TV, or listening to Radio, discussing the information about the plane crash with family and friends, etc.)” (Answer: “<1 hour per day”; “1-2 hours per day”; “≥3 hours per day”). We have discussed the impact of media coverage under the subtitle- 4.2. Experience and attitudes, showing as “In addition, we found that 37.2% of attendant students spent at least 1 hour per day ruminating about the plane crash, which is much higher than the percentage of the general sample (16.1%). This may be one reason for the higher prevalence of PTSS among the students. The media bring people convenient information, but they often prefer to provide negative news, which is often more eye-catching. Therefore, there is a need to enhance the proper understanding of aircraft accidents and avoid over-focusing on negative reports, thereby reducing the appearance of fear and negative emotions.”
